# Effectiveness Evaluation of *Viti’s vinifera* Leaf Extract on the Viability of Echinococcus Eggs and Protoscolices In Vitro

**DOI:** 10.3390/vetsci10060400

**Published:** 2023-06-18

**Authors:** Mohammed M. Mares, Saleh Al-Quraishy, Mutee Murshed

**Affiliations:** Department of Zoology, College of Science, King Saud University, P.O. Box 2455, Riyadh 11451, Saudi Arabia; mmares@ksu.edu.sa (M.M.M.);

**Keywords:** *Echinococcus*, hydatid cysts, carnivorous, *Vitis vinifera* leaf, sporicidal

## Abstract

**Simple Summary:**

Hydatidosis in humans and other animals represents a significant public health and economic problem with global prevalence. Adult tapeworms in dogs and a range of domesticated intermediate hosts, like cattle, sheep, goats, and camels, keep the parasite’s domestic life cycle going. Human infection may occur after ingesting infective eggs, passed in dogs’ stools through intimate contact. The extensive use of synthetic *E. granulosus* treatments can lead to resistance if used for an extended period at inappropriate doses, prompting the investigation of plant extracts and compounds derived from it with fewer side effects and which are cheaper. *Viti’s vinifera* leaf extracts are attractive potential candidates for alternative therapeutic approaches. Our results suggest that the *V. vinifera* leaf methanol extract reduces eggs and the protoscolices of *Echinococcus granulosus*.

**Abstract:**

Echinococcosis is a zoonotic disease caused by the genus Echinococcus. Globally, it is one of the most central helminthic diseases. Surgery remains the method of choice to remove cystic Echinococcus. Various sporicidal agents have been used to invalidate the substances in hydatid cysts. Nevertheless, many sporicidal agents cause inflammation and can cause side complications, so their use should be limited. The study aims to evaluate the effectiveness of Vitis vinifera leaf methanolic extract as a sporicidal agent for Echinococcus eggs and protoscolices and determines the best concentration. The mortality and viability of protoscolices were measured in samples exposed to four concentrations of V. vinifera leaf extract (VVLE) (5, 10, 30, and 50 mg/mL) for 5, 10, 20, and 30 min and in eggs exposed to three concentrations (100, 200, and 300 mg/mL) for 24 and 48 h. An infrared spectroscopy chemical test was conducted to assess the presence of numerous expected active components in the extract. The viability of eggs and protoscolices was confirmed using 0.1% eosin staining. Vinifera leaf extract exhibited the decisive sporicidal effect at 100%, 91%, 60%, and 41% after 30 min at concentrations of 50, 30, 10, and 5 mg/mL, and in eggs at 11% and 19% after 24 and 48 h at a concentration of 200 mg/mL, respectively. Increased incubation times and higher dosages often increase mortality. The results exhibited that *V. vinifera* is effective. This study confirmed that grape leaf extract has high sporicidal activity in vitro. However, more studies are required to determine the exact active chemical and its action mechanism and perform in vivo utilization to confirm these results.

## 1. Introduction

Hydatidiasis is a chronic infectious disease of medical and veterinary importance caused by the larval stage of the international parasite *Echinococcus granulosus*. Cystic echinococci disease in humans and other animals presents global health and economic threats [1]. This parasite is endemic to much of Asia, including the countries of East Asia, the former Soviet republics of central, northern China, and northeastern Siberia. The infection is universally known to be current in many parts of the eastern Mediterranean, in the Arab countries’ frontier of the Persian Gulf [2]. Furthermore, *E. granulosus* is found in vast areas of Europe [3].

Adult worms are present in the small intestines of dogs and sometimes in other predatory animals. Still, the larval stage can settle in various intermediate hosts such as livestock, pigs, horses, and humans [4]. Human infections can occur after ingesting infectious eggs transmitted through intimate contact with dog feces or environmental pollution [5]. The released embryo penetrates the intestinal wall and travels through the portal system, primarily to the liver (40–70%), lungs (30–40%), or other organs where hydatid cysts develop [6]. Cysts consist of two parasite-derived layers. An inner nucleated germ layer and an outer acellular laminate layer surrounded by a fibrous capsule produced by the host as a result of the host’s immune response. Cysts typically grow 1–5 cm in diameter per year, depending on the density of the host tissue [7]. The cyst produces millions of protoscolices that are released when a predator eats the organs of an infected intermediate host. Swallowed protoscolices mature into adult worms that attach to the proximal small intestine to complete their life cycle [8]. Liver cysts can cause pain in the upper abdomen, hepatomegaly, cholestasis, biliary cirrhosis, portal hypertension, ascites, and many other manifestations. Cysts can rupture into the peritoneal cavity, causing anaphylaxis and cholangitis. An abscess can form after bacterial infection of the cysts. Chronic cough, sputum production, shortness of breath, hemoptysis, pleurisy, and lung abscess are selective signs caused by lung cysts and possibly neurological disorders caused by cysts in the brain [9]. Cysts arise in the liver, lung, or both in over 90% of cases. Cysts have been reported (2–4%) in the kidney, spleen, peritoneal bore, skin, and muscles but rarely in the brain or heart [10].

Surgery is still the most effective treatment for hydatid disease in numerous parts of the world, including Saudi Arabia [11]. However, surgery poses a risk of intraoperative scoliosis shedding, which is the leading cause of repetition and multiple secondary hydatidoses [12]. Many pesticides inactivate cystic contents, including hypertonic saline, agnitrate, cetrimide, povidone-iodine, and ethanol. However, these agents are often associated with toxicity, liver necrosis, and methemoglobinemia [10,13,14]. Therefore, many studies aspire to develop new pesticides that are less harmful to patients, mainly from natural products, to treat hydatidosis [15,16].

Recent studies on plant extracts and compounds of plant origin demonstrate a valuable tactic for treating many diseases due to fewer side effects, little cost, and wide availability [17]. *V. vinifera* is an Asian native perennial woody vine. Several preparations have been prepared for use in traditional medicine from various parts of this plant, mainly the fruit and leaves [18]. It is wealthy in beneficial antioxidant compounds, including epicatechins, anthocyanins, catechins, and flavonoids [19]. Additionally, the aqueous extract of the leaf of *V. vinifera* reportedly exhibits antibacterial activity and treats *E. coli*, *Enterococcus feacalis*, and *Staphylococcus aureus* [20]. The grape seed extract is used against *Trichostrongylus colubriformis* and *Ostertagia circumcincta* in sheep [21]. It was used to treat diarrhea [22].

This study aimed to evaluate the in vitro scolicidal effects of the methanolic extract of *V. vinifera* leaf on the hydatid cyst protoscolices.

## 2. Materials and Methods

### 2.1. Preparation of Extract

The *V. vinifera* leaf was collected from a local Riyadh, Saudi Arabia market. A taxonomist from the Department of Botany at King Saud University confirmed the plant’s botanical identity. After air drying 500 g at 44 °C, the dried leaves were ground into a fine powder (particle size: 0.25–0.30 mm) with an electric mixer. Next, a magnetic stirrer added 120 g of dry powder to 400 mL of 70% methanol and gently blended for 1 h. The resulting solution was stirred vigorously at room temperature for 24 h with vigorous stirring and then filtered. The extract was concentrated and dried in a rotary vacuum evaporator (Yamato RE300, Tokyo, Japan). The section was resolved in distilled water for diverse investigations in experimental studies.

### 2.2. Infrared Spectroscopy

Some material was mixed with excess potassium bromide powder at 1:99% weight and processed uniformly. The material was then coarsely ground and put into a granulation die. The optical spectrometer NICOLET 6700 Fourier-transform infrared spectroscopy from Thermo Scientific (Waltham, MA, USA) was used to examine infrared (IR) (FT-IR) to determine the likelihood in the samples. The maximum absorbance is expressed in wave numbers (cm^−1^). Spectra ranging from 4000 cm^−1^ to 400 were registered at 25 °C with a resolution of 4 cm.

### 2.3. Collection of Dog Fecal Samples and Isolation of Eggs

Thirty stool samples were collected from various places in Al-Kharj (Saudi Arabia), where livestock rearing occurs in the presence of stray and domestic dogs. These samples were placed in tubes with capacities of 50 mL before being transferred to the parasitology laboratory at King Saud University’s Department of Zoology and kept at 4 °C until analyzed. After that, the parasite eggs were collected from the fecal samples using immediate wet mount microscopic examination and the sedimentation technique, followed by centrifugation at 1008× *g* for 10 min [23]. The eggs were removed from the coverslip with a 0.9% NaCl solution and kept at −20 °C.

### 2.4. Determination of In Vitro Effects on Eggs

The extract solutions were prepared at three concentrations (100, 200, and 300 mg/mL). In each experiment, 1 mL extracts of the suspension of eggs (containing at least 700 eggs) were placed in test tubes using a pipette. Then, 1 mL of various concentrations of the VVLE were added to separate test tubes, blended gently, and then incubated at room temperature for 24 and 48 h. At the end of each incubation period, 0.5 mL of the egg extract suspension was collected and mixed gently with 0.5 mL of 0.1% eosin dye. After 30 min, the residual settled eggs were spread on a slide, covered with a covered glass, and examined under a light microscope for viability. Viable eggs retained their color, but dead eggs turned red. The ratios of possible to dead eggs were determined by scoring at least 150 eggs. In addition, at least 700 eggs in 1 mL of distilled water were used as the control group.

### 2.5. Collection of Protoscolices

The protoscolices were collected from infected livers and lungs of animals that contained hydatid cysts slaughtered at the Al-Kharj slaughterhouse in Saudi Arabia and transferred to the parasitology laboratory of the zoology department of the University of King Saud. After opening cysts using a scalpel, all the liquid in the cysts was withdrawn using a pipette, placed in sterile bottles, and left to stand for 30 min to let the protoscolices gather at the bottom of the bottles. The supernatant was then discarded, and the yielded protoscolices were washed three times with ordinary sterile saline. A fertility test was performed, and the viability was assessed via muscular movement and a 0.1% eosin staining check under light microscopy. Lastly, live protoscolices were transferred to a dark container with ordinary sterile saline and kept at 4 °C for later use.

### 2.6. Determination of In Vitro Effects on Protoscolices

To investigate the scolicidal effects of *Vitis vinifera* leaf against protoscolices of *E. granulosus*, we used various concentrations (5, 10, 30, and 50 mg/mL) of the extract for 5, 10, 20, and 30 min. To prepare the diverse concentrations, 0.05, 0.1, 0.3, and 0.5 g of the dried extracts were dissolved in 10 mL of distilled water. In each concentration, 2 mL extracts of the protoscolices solution (containing at least 2700 protoscolices) were transferred to a test tube using pipettes. Then, 2 mL of VVLE at the different concentrations was added to all test tubes and mixed gently, followed by incubation at 37 °C for 5, 10, 20, and 30 min. At the end of every incubation period, the upper phase was removed carefully to avoid disturbing the settled protoscolices. Then, 100 μL of 0.1% eosin dye was added to the remaining settled proctoscope and mixed gently. After 5 min, the top part of the solution was again discarded. The settled protoscolices were smeared onto a glass slide, covered with a glass cover slip, and assessed microscopically for viability. In addition, at least 2700 protoscolices in 2 mL of distilled water were used as the control group.

### 2.7. Viability Test

To assess the viability of protoscolices, a 0.1% eosin solution (1 g of eosin powder dissolved in 1000 mL of distilled water) was used [23].

### 2.8. Statistical Analysis

One-way ANOVA was performed using a statistical package program (Sigma Plot version 11.0). All *p*-values were two-sided, and *p* ≤ 0.001 was considered significant.

## 3. Results

As shown in Figure 1, spectroscopy analysis using an FT-IR spectrometer was performed on extracts of *V. vinifera* leaf with influential bands at 3382.1 cm^−1^, 2119.8 cm^−1^, 1632.20 cm^−1^, 1510.70 cm^−1^, 1420.51 cm^−1^, and 1030.32 cm^−1^, respectively. The active phytochemicals obtained are as follows: N–H stretch, N=C=S stretch, C=C stretch, C–O tertiary stretch, CO–O–CO stretch, and C-H bend, with the absorbance at 400–4000/cm^−1^ (Table 1).

The viable eggs did not change color; dead eggs were stained red (Figure 2). The scolicidal effects of different VVLE concentrations are summarized in Figure 3 and Figure 4. The results obtained from the viable versus nonviable rates of *E. granulosus* eggs using *V. vinifera* leaf extract, following treatments for 24 h with concentrations of 100, 200, and 300 mg/mL, nonviable rates of 2.6%, 11%, and 8%, were observed, respectively. With the increased treatment periods of 48 h, nonviable rates of 6%, 19%, and 13% were observed on the same concentrations of VVLE. The low concentration (100 mg/mL) was significantly more lethal than the control (distilled water) at 24 and 48 h of exposure. 

Incubation for 30 min with concentrations of 30 and 50 mg/mL of VVLE led to death rates of 93% and 100%, respectively, with the dissolution and rupture of the protoscolices walls and cracking of the hooks (Figure 5).

The scolicidal effects of protoscolices at several concentrations of V. vinifera leaf extract are summarized. After incubation with concentrations of 30 and 50 mg/mL of VVLE, mortality rates of 40% and 60% were observed following treatment periods of 5 min compared to the control group, which did not exceed a mortality rate of 5%. However, at doses of 5 and 10 mg/mL, low mortality rates were observed following treatment for the same period (Figure 6).

After incubation with concentrations of 30 and 50 mg/mL of VVLE, mortality rates of approximately 70% and 75%, respectively, were observed for the protoscolices following treatment periods of 10 min compared to the control group, which had a death rate of 8%. However, at concentrations of 5 and 10 mg/mL of VVLE, 22% and 31% mortality rates were observed following treatment for the same period (Figure 7).

After incubation for 20 min with concentrations of 30 and 50 mg/mL of VVLE, the mortality rate of the protoscolices was higher than the previous ones after treatment, at approximately 83% and 95%, respectively. However, the death rate increased in the 5 and 10 mg/mL concentrations with the increase in the duration under treatment when it reached 50%. The control group had a mortality rate of 10% (Figure 8).

However, the death rate increased in the 5 and 10 mg/mL concentrations with the increase in the duration under treatment when it reached 63%. The control group had a mortality rate of 14% (Figure 9).

Figure 10 and Figure 11 depict the significant effects of the mortality rate and experimental groups on the viability and death of protoscolices in vitro. The mortality proportion increased with increasing incubation times and, conversely, the viability percentage. The mortality of the protoscolices significantly increased with incubation times up to 20 min (*p* < 0.05), thus indicating quite different viability between 5, 10, and 30 min exposure.

It shows the protoscolices’ viability proportion on VVLE as a function of the concentrations and treatment times. The figure demonstrates that all times, an increase in the doses seems to enhance its activity. As a result, as the concentration increased, so did the mortality rates. The VVLE, so, has the potential to be implemented better at 50 mg/mL and probably at higher concentrations. As shown in Figure 9, after 30 min, VVLE at various concentrations determined the viability of protoscolices in a concentration-dependent manner compared to the control groups. Most engagements include anti-protoscolices activities at 10, 30, and 50 g/mL. The highest death rate percentage was 100% at a dose of 50 mg/mL of VVLE toward the protoscolices strain. The lowest efficacy toward the protoscolices was 12% at a 5 mg/mL concentration after 5 min (Figure 12).

The differences in the mortalities because of the leaf extract of *V. vinifera* were statistically highly significant (*p* < 0.01) for the 50 mg/mL concentration at 20–30 min application times.

## 4. Discussion

Despite better control of hydatidosis, this zoonotic disease remains a major public health problem in some regions [24]. The yearly incidence of cystic echinococcosis ranges from <1 to 200 per 100,000 inhabitants in endemic regions, that of alveolar echinococcosis ranges from 0.03 to 1.2 per 100,000 population [25] but may be significantly higher in certain endemic foci (WHO/OIE, 2001). In untreated cases, mortality is >90% within 10–15 years of diagnosis [26]. Surgery is a suitable treatment for particular cases. Chemotherapy with benzimidazole and PAIR (puncture, suction, insertion, and ventilation) is recommended as an alternative, chiefly for patients who cannot endure surgery [27]. The main surgical problems of cystic echinococcosis are relapse, subaltern cystic echinococcosis, and anaphylactic shock because of intraoperative rupture of the cyst and leakage of the contents of the cyst (protoscolices), which is noticed in almost 10% of infected cases [13,28]. Therefore, many protocol side agents have been used, such as hypertonic saline, alcohol, and povidone-iodine [14]. Many scolicidal agents have been used to inactivate the contents of *Echinococcus multilocular* cysts, but none are effective and safe [10]. The use of plant extracts on protoscolices has recently garnered much interest. Some research showed extracts of plant types might also affect the viability of protoscolices [29]. This study explored the protoscolicidal effects of *V. vinifera* methanol extracts for use on protoscolices in hydatid cysts. *V. vinifera*-leaf-based medicines are traditionally used for diarrhea, hepatitis, and stomach aches [10]. Additionally, the extract of *V. vinifera* leaf reportedly exhibits antibacterial activity on *E. coli*, *Enterococcus feacalis*, and *Staphylococcus aureus* [20]. Water and methanol extracts of the seeds and leaf of *V. vinifera* have also been proven to have antibacterial activity against *Candida albicans, Candida glabrata*, and *Candida tropicalis* [30,31], summarize that grape seed and organic zinc improved the harm on the growth performance, damage score, and oocyst shedding (slightly) in chickens affected by *E. tenella*. The control of hydatidosis by inactivating *E. granulosus* eggs is effective. This study showed that all dilutions of (VVE) considerably increased the number of eggs killed compared to the control (distilled water). The high potency of *V. vinifera* extract may be due to the activity of its antioxidant compounds against *Echinococcus* eggs in vitro including flavonoids, anthocyanins, catechin, and epicatechins [32]. These results are similar to [26] findings that *V. vinifera* extract had a curbed dose-dependent effect on the sporulation and harm of *Eimeria* oocyst. Among all the tested concentrations, *V. vinifera* extract at a 200 mg/mL dose significantly increased the number of eggs killed compared to other concentrations, which may be due to the presence of substances that affect the mortality of eggs with a concentration of 200 more than the doses of 100 and 300 mg/mL. The authors of [4] tested the methanolic extract of *V. vinifera* as anti-*leech limnatis nilotica* in vitro and found that the methanol extract of grapes could be presented as a complementary treatment against *leech limnatis nilotica*. The results showed that *V. vinifera* leaf extract elicited a strong sporicidal effect at all doses and exposure times; similarly, we observed 96.7% and 100% death rates at concentrations of 50 mg/mL of methanolic extracts of *V. verifier* leaf after 20 and 30 min of exposure, respectively. This means that concentrations of 50 mg/mL have larger scolicidal effects in a low exposure time than the other concentrations. The lower doses (5 mg/mL) were significantly more lethal than the negative control at all exposure times. These concentrations used for the *V. vinifera* leaf extract are considered low and very effective protoscolicidal agents compared to the diverse plants that achieved the same goal. The power of scolicidal activity against protoscolices varies greatly between extracts from different plant species. For example, *Nectaroscordum tripedal* extracts at 100 mg/mL after 5 min of exposure killed 100% of the protoscolices [33]. The authors of [34] reported that a 100% protoscolices mortality was reached after using 300 and 150 mg/mL of *Olea Europaea* leaf extract for 10 and 20 min of incubation, respectively. This study’s scolicidal effect at 50 mg/mL caused the protoscolices walls to disintegrate and rupture and lose the hooks. The authors of [20] found out that the aqueous and ethanolic extract of *V. vinifera* promoted the disintegration of the nuclear and cytoplasmic membrane of *Leishmania infant* promastigote and changed the cells’ overall shapes. This may be because of the grape extract containing alcohol, which affects the formation of the protocol wall. The authors of [35,36] reported that alcohol alters the cell membranes and affects several types of stem cells, including embryonic, liver, intestinal, bone-marrow-derived stromal cells, and neurons cells. However, this study is the initial to demonstrate the scolicidal activity of the methanolic *V. verifier* leaf extract.

## 5. Conclusions

Our results suggest that the *V. vinifera* leaf methanol extract affects the outer wall of both eggs and protoscolices of *E. granulosus*, which may affect their vitality in vitro. Further, in vivo studies are necessary to evaluate the potency of this extract or some of its purified components as a beneficial alternative for the therapy of echinococcosis.

## Figures and Tables

**Figure 1 vetsci-10-00400-f001:**
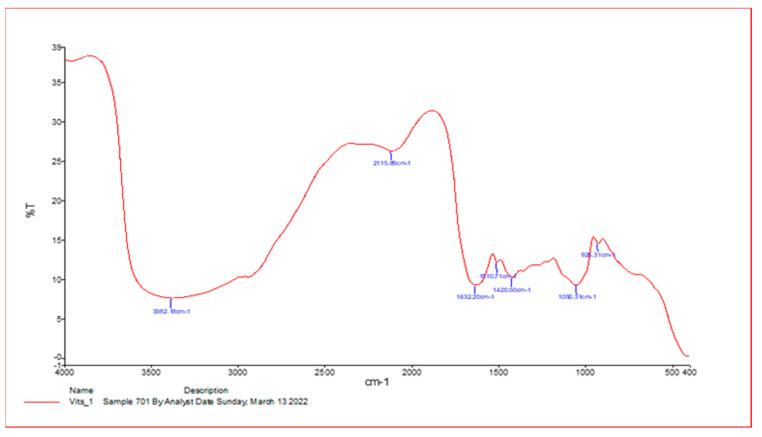
Infrared spectroscopy results from *Vitis vinifera* leaf extract samples. An FT-IR spectrometer was used for data collection and provided results ranging from 400 to 4000/cm^−1^.

**Figure 2 vetsci-10-00400-f002:**
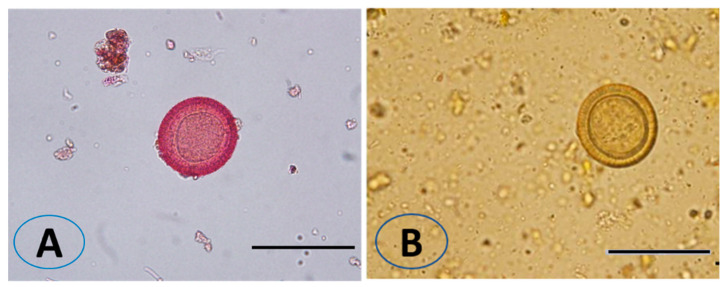
Effect of *Vitis vinifera* leaves extract on eggs viability. The unviable eggs after treatment with extract and staining with 0.1% eosin (**A**). The viable eggs did not change color after staining with 0.1% eosin (**B**). Scale bar = 20 µm.

**Figure 3 vetsci-10-00400-f003:**
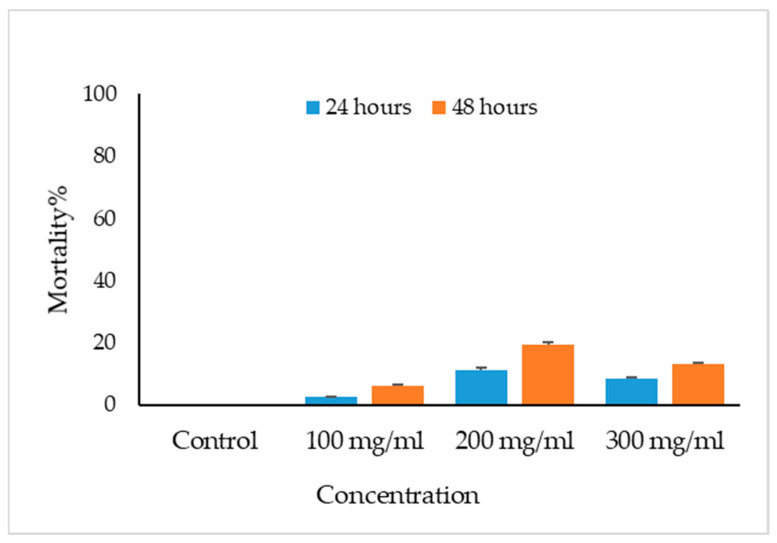
Main effects of *Vitis vinifera* leaf extract on mortality rates of eggs at different concentrations of contact time, and treatment effects at 100, 200, and 300 mg/mL in vitro.

**Figure 4 vetsci-10-00400-f004:**
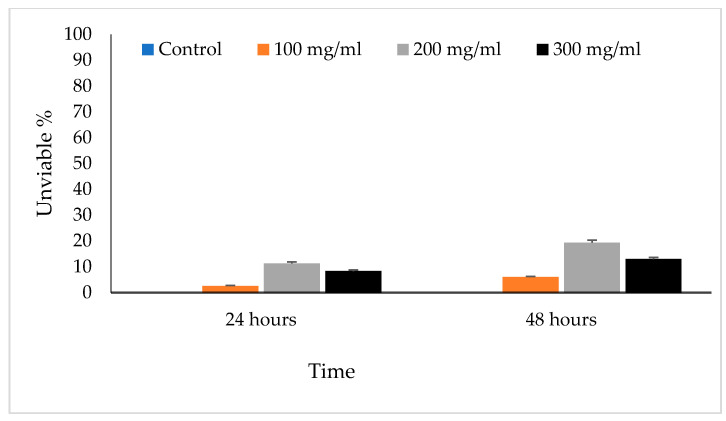
The main effects of *Vitis vinifera* leaf extract on viable eggs at 24 and 48 h of exposure and treatment effects at different concentrations in vitro.

**Figure 5 vetsci-10-00400-f005:**
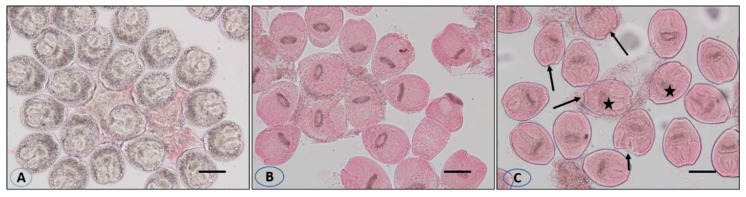
Live protoscolices after staining with 0.1% eosin (**A**), dead protoscolices after treatment with the extract and staining with 0.1% eosin (**B**), and dissolution and rupture of the protoscolices wall (arrows) with lack of hooks (asterisk) (**C**). Scale bar = 10 µm.

**Figure 6 vetsci-10-00400-f006:**
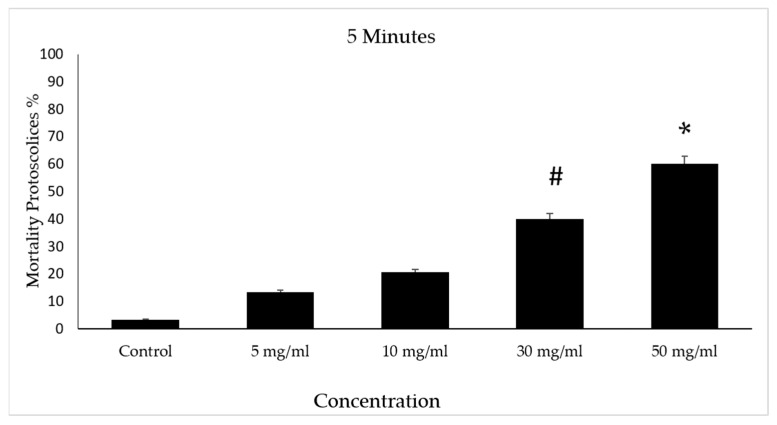
Effect of *Vitis vinifera* leaves extracts on protoscolices mortality at 5 min in vitro. Significance (#): *p*-value ≤ 0.05 and (*): *p*-value ≤ 0.01.

**Figure 7 vetsci-10-00400-f007:**
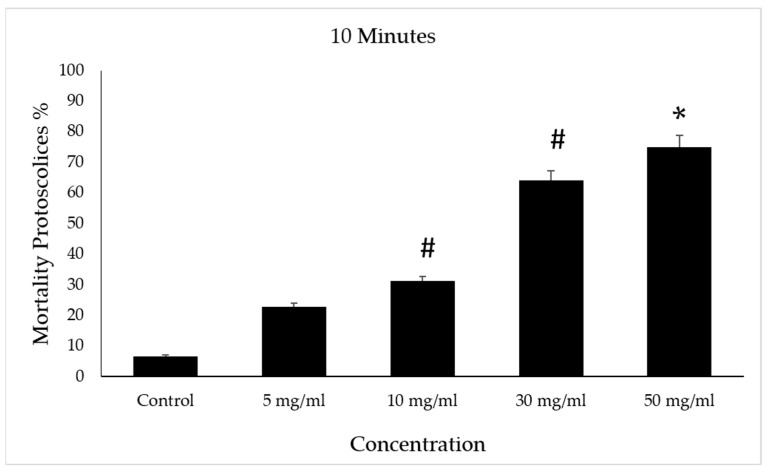
Effect of *Vitis vinifera* leaves extracts on protoscolices mortality at 10 min in vitro. Significance (#): *p*-value ≤ 0.05 and (*): *p*-value ≤ 0.01.

**Figure 8 vetsci-10-00400-f008:**
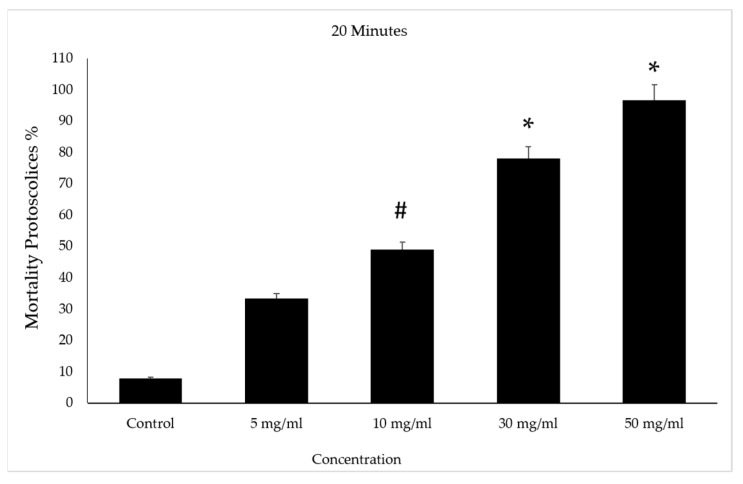
Effect of *Vitis vinifera* leaves extracts on protoscolices mortality at 30 min in vitro. Significance (#): *p*-value ≤ 0.05 and (*): *p*-value ≤ 0.01.

**Figure 9 vetsci-10-00400-f009:**
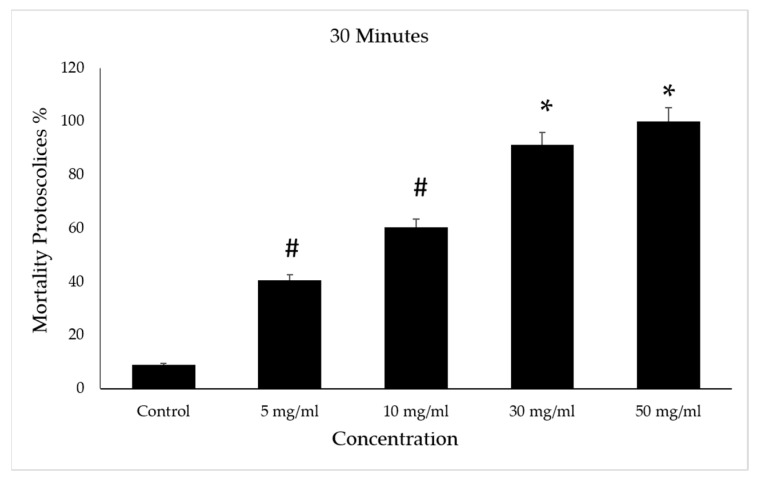
Effect of *Vitis vinifera* leaves extracts on protoscolices mortality at 50 min in vitro. Significance (#): *p*-value ≤ 0.05 and (*): *p*-value ≤ 0.01.

**Figure 10 vetsci-10-00400-f010:**
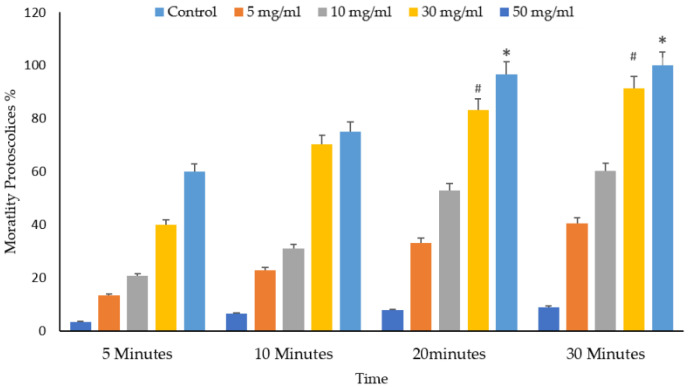
Main effects of *Vitis vinifera* leaf extract on mortality rates of protoscolices at different doses of extract of varying contact time and treatment effects at 5, 10, 20, and 30 min in vitro. The significance was compared to distilled water as a control. Statistical significance in comparison to the control group. (*): indicates *p*-value ≤ 0.01 and (#): indicates *p*-value ≤ 0.05.

**Figure 11 vetsci-10-00400-f011:**
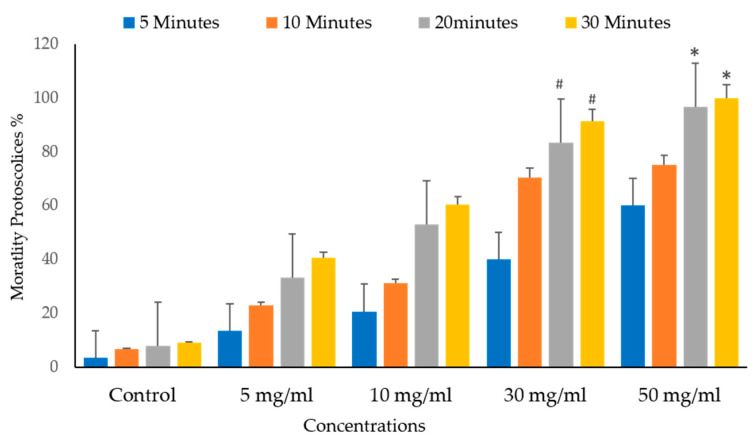
Main effects of *Vitis vinifera* leaf extract on mortality rates of protoscolices at different times of varying contact start and treatment effects at 5, 10, 20, and 30 mg/mL in vitro. The significance was compared to distilled water as a control. Statistical significance in comparison to the control group. (*): indicates *p*-value ≤ 0.01 and (#): indicates *p*-value ≤ 0.05.

**Figure 12 vetsci-10-00400-f012:**
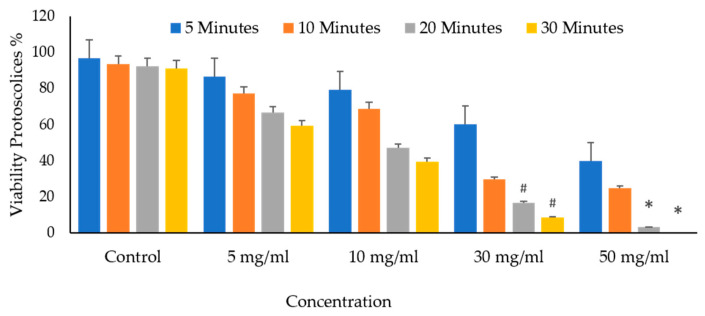
In vitro effects of *Vitis vinifera* leaf extract on protoscolices viability at various concentrations after 5, 10, 20, and 30 min. The significance was compared to distilled water as a negative control. (*) indicates a *p*-value of ≤ 0.01, while (#) indicates a *p*-value of ≤ 0.05.

**Table 1 vetsci-10-00400-t001:** The IR spectrum of *V. vinifera* leaf extracts by frequency range.

Absorption (cm^−1^)	Appearance	Transmittance (%)	Groups	Compound Class
3425.5	Medium	12	N-H stretch	Aliphatic primary amine
2093.1	Strong	47	N=C=S stretch	Isothiocyanate
1641.4	Strong	25	C=C stretch	Alkene
1209.1	Strong	37	C-O stretching tertiary	Alcohol
1045.6	Strong broad	35	CO-O-CO stretch	Anhydride
410.4	Strong	3	C-H bend	1,2-disubtituted

## Data Availability

Not applicable.

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
