# Peer review of "Effectiveness Evaluation of Viti’s vinifera Leaf Extract on the Viability of Echinococcus Eggs and Protoscolices In Vitro"

_vetsci, 2023, doi:10.3390/vetsci10060400_

Round 1

Reviewer 1 Report

Please refer to attached Word document for detailed comments and suggestions.

Author Response

Dear Editor

 Dear Reviewer

I would like to thank you very much for your kind review of my manuscript and for showing it in a good appearance entitled (Effectiveness evaluation of Vitis vinifera leaf extract on the Vi-ability of Echinococcus eggs and Protoscolices: In Vitro).

I answered all the questions and mandatory requirements in detail step by step marked in yellow

Commented [A1]: Cattle, sheep, goats, and camels are not domestic pets as they do not generally live in houses.

Done

Commented [A2]: There seems no justification or benefit to reporting the results with accuracies to 2 decimal places.

Done

Commented [A3]: These are all signs, not symptoms.

Done

Commented [A4]: Citation 22 is titled "In vitro anticoccidial activity of Vitis vinifera extract on oocysts of different Eimeria species of broiler chicken" and does not appear to deal with diarrhea. Also, treating a disease with an agent does not mean it is effective. Was it shown to be effective?

Correct reference added

J.D. Felicio, R.S. Santos, E. Goncalez, Chemical constituents from Vitisvinifera (Vitaceae), Arq. Inst. Biol. 68 (2001) 47–50.

Commented [A5]: To what concentration? The obtained extract was concentrated and dried in a rotary vacuum evaporator (Yamato RE300, Japan).

The concentration here means the removal of water and methanol from the extract in the evaporator

Commented [A6]: How much? Amounts should be exact.

0.001% sample

Commented [A7]: How much? Amounts should be exact.

99.999% potassium bromide powder

Commented [A8]: Were they collected randomly or were the dogs selected? If the latter, what were the selection criteria? Same for the collection sites. Were they random or selected?

Soft stool samples were randomly collected from random places in Al-Kharj city and also near places where dead animal corpses were thrown.

Commented [A9]: That means the mass of the samples must have been measured. This needs to be described above. How many replicates were made at each concentration?

It was not mentioned in the above paragraph, because it talks about the processing of the extract from the papers until the extract is ready.

As for the concentrations used, three concentrations were used to test the eggs, where 100 mg of the solid extract was dissolved in one ml of distilled water, and so on in proportion to the rest of the concentrations.

Commented [A10]: How did you determine the red-stained eggs were nonviable? Can you cite an accepted source for this? What colour were the viable eggs?

We note in the attached picture in the paper that the outer layer of the eggs is irregular and we note that the red color inside the eggs, even the size of the eggs is larger than the eggs that are not dyed red, and this is evidence of the effect of the eggs.

As for the color of the eggs, we note that the natural color of the eggs is brown, as is the color in the other picture attached to the paper

This reference shows what nonviable eggs looked like and how they were stained red

Moazeni, M., & Rakhshandehroo, E. (2012). In vitro viability test for the eggs of Echinococcus granulosus: a rapid method. Parasitology Research, 110, 925-930.‏

Commented [A11]: How many control groups were there and for which test groups?

One set was used but there was a typing error and it has been modified. control groups.  control group.

Commented [A12]: You must describe how this was done. Was the surface seared with a scalding metal knife before cutting out the sample with a sterile scalpel blade and transferring it to a sterile Petri dish?

Organs affected by the hydatid cysts were collected and placed in bags and placed in refrigerated containers and transported to the laboratory, and the heads were collected in the laboratory after opening the water bags as shown in the paper.

Also, we used another method if the organs were infected with a few hydatid cysts with one cyst, for example, the hydatid cyst was removed from the affected organ using a knife and kept in tubes of 50 ml capacity

Commented [A13]: As above you need to describe in detail how this was done so other researchers can repeat the process.

has been added to the paper

after opening cysts using a scalpel, all the liquid in the cysts was withdrawn using a pipette and placed in sterile bottles, and left to stand for 30 min; the protoscoleces sank to the bottom of the bottles.

Commented [A14]: You need to explain why this was done. Was it to allow the protoscoleces to sink to the bottom for collection?

has been added to the paper

and left to stand for 30 min; this is to let the protoscoleces gather at the bottom of the bottles.

Commented [A15]: How were they modified?

The word modified has been deleted

No modification was done, only the extract was dissolved in distilled water

Commented [A16]: Most research projects use a 5% level of significance (P< 0.05), and far fewer use 1% (P< 0.01). Why did you use such a high level of significance (0.1%, P < 0.001)?

P < 0.001, it was correcting

P< 0.01 indicates high significant differences, and this is what happened between the control and the highest concentration. While P< 0.05 indicates the presence of close or low significant differences between the low concentrations or with the control.

Commented [A17]: You should explain in a "Key:" what is meant by "Appearance" and the differences between "Medium, "Strong" and "Strong broad". You should also describe what you mean by "Transmittance".

Medium, Strong, and Strong broad This expresses the number of active chemicals in the extract. Some compounds may be present in greater quantity than others. while "Appearance" expresses the presence of active chemicals in the extract

Commented [A18]: Accuracy to two decimal places does not seem to be justified or useful.

Done

Commented [A19]: You should state the levels of significance for both exposures.

 because the death rate was very low

Commented [A20]: The eggs in both A and B images are stained red. What colour is the egg in A supposed to be?

In picture A, the eggs were not dyed in red color, as the egg wall was not dyed and was not affected compared to the other picture.

We note in the attached picture in the paper that the outer layer of the eggs is irregular and we note that the red color inside the eggs, even the size of the eggs is larger than the eggs that are not dyed red, and this is evidence of the effect of the eggs.

As for the color of the eggs, we note that the natural color of the eggs is brown, as is the color in the other picture attached to the paper

Commented [A21]: The mortalities with the different concentrations and two-time exposures should include the control mortalities. How many controls were used?

During the examination period, no eggs died in the control group.

One control group was used

It was not added to the drawing, because the death rate zero

Commented [A22]: Don't you mean on viable eggs?

I mean unviable eggs

Commented [A23]: The control results should be included in the figure. How many controls were used?

One control group was used

It was not added to the figure, because the death rate was very low, but this was mentioned in the explanation about the control group in terms of eggs

Commented [A24]: The mortality rates should be expressed to the same level of accuracy. Accuracy to 2 decimal places seems unjustified and even one decimal place

seems suspect.

Done

Commented [A25]: A result of 70.33% is supposedly accurate to 2 decimal places and cannot be described as "approximately". The result should be writen as 70%. The same logic should be applied to all the results.

Done

Commented [A26]: You switch from "mortality rate" to "death rate". You should be consistent.

Done

Commented [A27]: I do not know what this means and cannot find a meaning for it online.

The sentence in the paper has been modified

Commented [A28]: You have to explain any acronym the first time you use it. Does CE stand for "cystic echinococcosis"? Does AE stand for "alveolar echinococcosis"?

Done

CE  cystic echinococcosis

AE alveolar echinococcosis

Commented [A29]: What does this mean? The main surgical multiples of cystic echinococcosis are relapsed, subaltern cystic echinococcosis, and anaphylactic shock

That is, a secondary infection occurs when the water sacs are opened and this fluid that contains the heads leaks to other Organ, and therefore the surgery is considered dangerous in the occurrence of a secondary infection. Also, surgery and opening the sacs cause great complications for those infected with it.

Commented [A30]: What does this mean?

That is, a secondary infection occurs when the water sacs are opened and this fluid that contains the heads leaks to other Organs, and therefore the surgery is considered dangerous in the occurrence of a secondary infection. Also, surgery and opening the sacs cause great complications for those infected with it.

Commented [A31]: I don't know what this means here.

The sentence in the paper has been modified

Commented [A32]: What does this mean?

The sentence in the paper has been modified

Commented [A33]: There seems to be something missing here. It does not make sense.

Note that the grape plant is effective against eggs and protoscolicidal compared to plants that were used against eggs and protoscolicidal, and this is what was meant by this paragraph.

Commented [A34]: What does this mean?

Note that the grape plant is effective against eggs and protoscolicidal compared to plants that were used against eggs and protoscolicidal, and this is what was meant by this paragraph.

Reviewer 2 Report

The objective and experimental design are interesting; however the quality of English language is reducing drastically the value of the paper. Authors should review English language in all the paper and once given, abbreviations should be used instead of full definition. 

Also, figure titles should be accurately written, and abbreviation should be avoided unless full definition is given in the description. when comparison has been done, authors must provide what are the groups of comparison. Axis labeling should be clear with full details. Saying "concentration" is not full description! (Concentration of what?)

Other remarks are in the attached PDF.

Good luck. 

Author Response

Dear Editor

 Dear Reviewer

I would like to thank you very much for your kind review of my manuscript and for showing it in a good appearance entitled (Effectiveness evaluation of Vitis vinifera leaf extract on the Vi-ability of Echinococcus eggs and Protoscolices: In Vitro).

I answered all the questions and mandatory requirements in detail step by step marked in yellow

what material?

What is meant by materials, which is extract from Vitis vinifera leaf

what do you mean? surplus

It has been changed to the word “excess” and it means that the required amount of extract is mixed with potassium bromide in an amount twice the amount of the extract

Please rewrite!

Done

The optical spectrometer NICOLET 6700 Fourier-transform infrared spectroscopy from Thermo Scientific was used to examine infrared (IR) (FT-IR), to predict the likely compound classes.

review text font

Done

In vitro should be in italic in all the paper

Done

species name must be in italic

Done

what do you mean ?

The word kindly has been replaced by the word gently, which is intended to gently mix the protoscoleces with the concentrations extract

remove italic

Done

once abbreviation was given you cannot use full definition anymore!

Done

give the figure title before giving the brief description

Done

Effect of Vitis vinifera leaves extract on eggs viability

the significance is not shown in the figure!

Done

the significance is not shown in the figure!

Done

use the full definition in titles of figures

Done

significance of what and compared to what? The same thing for the other figures.

Also say clearly the concentration of what, in the axis?

Comparison between extract concentrations and their effect on protoscolices

be consistent with the use of colors in all the figures.

The color has been changed in the figure to show the differences between several variables and to compare them

say how it affects?

Done

     Our results suggest that the V. vinifera leaf methanol extract affects the outer wall of both eggs and protoscolices of E. granulosus, which may affect their vitality in vitro. Further in vivo studies are necessary to further evaluate the potency of this extract or some of its purified components as a beneficial alternative for the therapy of echinococcosis.

Abbreviation

Done

Italic in vivo

Done

Round 2

Reviewer 1 Report

Commented [A8]: Were they collected randomly or were the dogs selected? If the latter, what were the selection criteria? Same for the collection sites. Were they random or selected?

Authors’ reply: “Soft stool samples were randomly collected from random places in Al-Kharj city and also near places where dead animal corpses were thrown.”

This indicates 30 stool samples were collected and not from 30 dogs so the original wording should be changed to say that: “Thirty canine fecal samples were collected randomly from various places in Al-Kharj.”

Commented [A11]: How many control groups were there and for which test groups?

One set was used but there was a typing error and it has been modified. control groups.  control group.

I do not understand this response.

Commented [A12]: You must describe how this was done. Was the surface seared with a scalding metal knife before cutting out the sample with a sterile scalpel blade and transferring it to a sterile Petri dish?

Organs affected by the hydatid cysts were collected and placed in bags and placed in refrigerated containers and transported to the laboratory, and the heads were collected in the laboratory after opening the water bags as shown in the paper.

Also, we used another method if the organs were infected with a few hydatid cysts with one cyst, for example, the hydatid cyst was removed from the affected organ using a knife and kept in tubes of 50 ml capacity

In the paper the authors state: “The protoscoleces were aseptically collected from infected liver and lung ..” Their reply above does not explain how the samples were collected ‘aseptically’.

Commented [A16]: Most research projects use a 5% level of significance (P< 0.05), and far fewer use 1% (P< 0.01). Why did you use such a high level of significance (0.1%, P < 0.001)?

P < 0.001, it was correcting

P< 0.01 indicates high significant differences, and this is what happened between the control and the highest concentration. While P< 0.05 indicates the presence of close or low significant differences between the low concentrations or with the control.

This is in the ‘Materials and Methods’ section so it seems the authors chose the high level of significance (P< 0.001). I advise moving it to the ‘Results’ section to avoid confusion.

Commented [A17]: You should explain in a "Key:" what is meant by "Appearance" and the differences between "Medium, "Strong" and "Strong broad". You should also describe what you mean by "Transmittance".

Medium, Strong, and Strong broad This expresses the number of active chemicals in the extract. Some compounds may be present in greater quantity than others. while "Appearance" expresses the presence of active chemicals in the extract

Does this response indicate the authors will include these explanations in the paper?

Commented [A19]: You should state the levels of significance for both exposures.

 because the death rate was very low

This response does not report the P levels for the exposures.

Commented [A20]: The eggs in both A and B images are stained red. What colour is the egg in A supposed to be?

In picture A, the eggs were not dyed in red color, as the egg wall was not dyed and was not affected compared to the other picture.

We note in the attached picture in the paper that the outer layer of the eggs is irregular and we note that the red color inside the eggs, even the size of the eggs is larger than the eggs that are not dyed red, and this is evidence of the effect of the eggs.

As for the color of the eggs, we note that the natural color of the eggs is brown, as is the color in the other picture attached to the paper

As both eggs in Figure 2 A and B are red and the authors state the natural colour is brown, the authors’ response does not make sense to me.

Commented [A22]: Don't you mean on viable eggs?

I mean unviable eggs

This refers to: “Line 210: Figure 4. The main effects of VVLE on unviable of eggs at 24 and 48 hours of exposure ..” As the project is investigating how effective VVLE is in killing Echinococcus eggs they must be alive (viable) before exposure. If they are already dead (“unviable”) there is no point to the exposure.

Commented [A29]: What does this mean? The main surgical multiples of cystic echinococcosis are relapsed, subaltern cystic echinococcosis, and anaphylactic shock

That is, a secondary infection occurs when the water sacs are opened and this fluid that contains the heads leaks to other Organ, and therefore the surgery is considered dangerous in the occurrence of a secondary infection. Also, surgery and opening the sacs cause great complications for those infected with it.

Accepted, but I do not understand the use of “surgical multiples” or “subaltern” here.

The authors replied to several comments that I do not understand what they meant with “The sentence in the paper has been modified.” I do not know how the paper was modified so I cannot determine whether it is now any clearer to readers.

Author Response

I would like to thank you very much for your kind review of my manuscript and for showing it in a good appearance

Commented [A8]: Were they collected randomly or were the dogs selected? If the latter, what were the selection criteria? Same for the collection sites. Were they random or selected?

Authors’ reply: “Soft stool samples were randomly collected from random places in Al-Kharj city and also near places where dead animal corpses were thrown.”

This indicates 30 stool samples were collected and not from 30 dogs so the original wording should be changed to say that: “Thirty canine fecal samples were collected randomly from various places in Al-Kharj.”

Done

Commented [A11]: How many control groups were there and for which test groups?

One set was used but there was a typing error and it has been modified. control groups.  control group.

I do not understand this response.

Only one control group containing 700 eggs in distilled water was used

Commented [A12]: You must describe how this was done. Was the surface seared with a scalding metal knife before cutting out the sample with a sterile scalpel blade and transferring it to a sterile Petri dish?

Organs affected by the hydatid cysts were collected and placed in bags and placed in refrigerated containers and transported to the laboratory, and the heads were collected in the laboratory after opening the water bags as shown in the paper.

Also, we used another method if the organs were infected with a few hydatid cysts with one cyst, for example, the hydatid cyst was removed from the affected organ using a knife and kept in tubes of 50 ml capacity

In the paper the authors state: “The protoscoleces were aseptically collected from infected liver and lung ..” Their reply above does not explain how the samples were collected ‘aseptically’.

Aseptically: It was deleted from the paragraph and the method is explained in an easier way

It means sterile method, that is, all tools used in the process of collecting and transporting samples were clean and sterilized with 70% ethanol before use, in order to not contaminate the carcasses during sampling.

Commented [A16]: Most research projects use a 5% level of significance (P< 0.05), and far fewer use 1% (P< 0.01). Why did you use such a high level of significance (0.05%, P < 0.05)?

P < 0.05, it was correcting

P< 0.01 indicates high significant differences, and this is what happened between the control and the highest concentration. While P< 0.05 indicates the presence of close or low significant differences between the low concentrations or with the control.

Done

This is in the ‘Materials and Methods’ section so it seems the authors chose the high level of significance (P< 0.001). I advise moving it to the ‘Results’ section to avoid confusion.

Done

Commented [A17]: You should explain in a "Key:" what is meant by "Appearance" and the differences between "Medium, "Strong" and "Strong broad". You should also describe what you mean by "Transmittance".

Medium, Strong, and Strong broad This expresses the number of active chemicals in the extract. Some compounds may be present in greater quantity than others. while "Appearance" expresses the presence of active chemicals in the extract

Does this response indicate the authors will include these explanations in the paper?

I think it is not necessary to add this clarification in the paper because the method is known and clear

Commented [A19]: You should state the levels of significance for both exposures.

 because the death rate was very low

This response does not report the P levels for the exposures

Commented [A20]: The eggs in both A and B images are stained red. What colour is the egg in A supposed to be?

In picture A, the eggs were not dyed in red color, as the egg wall was not dyed and was not affected compared to the other picture.

We note in the attached picture in the paper that the outer layer of the eggs is irregular and we note that the red color inside the eggs, even the size of the eggs is larger than the eggs that are not dyed red, and this is evidence of the effect of the eggs.

As for the color of the eggs, we note that the natural color of the eggs is brown, as is the color in the other picture attached to the paper

As both eggs in Figure 2 A and B are red and the authors state the natural colour is brown, the authors’ response does not make sense to me.

There is an obvious difference in color, shape, and changes between picture A and picture B.

I will attach in the report a picture of the eggs before the test, which shows the natural color of the eggs.

We notice in the picture added in the report that it resembles the external shape of the viable egg A, where we notice in picture B that the dye has entered the inside of the egg and changes have occurred on the outer surface of the egg and its size has increased compared to the picture added to the report and picture A

Commented [A22]: Don't you mean on viable eggs?

I mean unviable eggs

This refers to: “Line 210: Figure 4. The main effects of VVLE on unviable of eggs at 24 and 48 hours of exposure ..” As the project is investigating how effective VVLE is in killing Echinococcus eggs they must be alive (viable) before exposure. If they are already dead (“unviable”) there is no point to the exposure.

The figure shows the effect of the extract on viable eggs and the amount of mortality in them after exposure to the extract and its effect on them to make them non-viable eggs.

Yes, in the experiment, viable eggs were used, not non-viable eggs

It was modified in the scientific paper from non-viable eggs to viable eggs

Commented [A29]: What does this mean? The main surgical multiples of cystic echinococcosis are relapsed, subaltern cystic echinococcosis, and anaphylactic shock

That is, a secondary infection occurs when the water sacs are opened and this fluid that contains the heads leaks to other Organ, and therefore the surgery is considered dangerous in the occurrence of a secondary infection. Also, surgery and opening the sacs cause great complications for those infected with it.

Accepted, but I do not understand the use of “surgical multiples” or “subaltern” here.

It means that if the liquid in the bags containing the protoscolios leaks, the infection will occur again, and in this way the infection continues in the animal, and this is considered one of the most important complications. Therefore, surgery is a problem, and it is preferable to use medicines or natural products that work to get rid of this parasite

The authors replied to several comments that I do not understand what they meant with “The sentence in the paper has been modified.” I do not know how the paper was modified so I cannot determine whether it is now any clearer to readers

Round 3

Author Response

I would like to thank you very much for your kind review of my manuscript and for showing it in a good appearance. Your valuable comments added to my manuscript a lot of clarity. Thank you again, reviewer.

Commented [A8]: Were they collected randomly or were the dogs selected? If the latter, what were

the selection criteria? Same for the collection sites. Were they random or selected?

Authors’ reply: “Soft stool samples were randomly collected from random places in Al-Kharj city and also near places where dead animal corpses were thrown.”

This indicates 30 stool samples were collected and not from 30 dogs so the original wording should

be changed to say that: “Thirty canine fecal samples were collected randomly from various places in Al-Kharj.”

Changes were made to the manuscript as per your directions.

30 stool samples were collected from various places in Al-Kharj (Saudi Arabia) where livestock rearing occurs in the presence of stray and domestic dogs.

Done ✓

Commented [A11]: How many control groups were there and for which test groups?

One set was used but there was a typing error and it has been modified. control groups. control

group.

I do not understand this response.

Only one control group containing 700 eggs in distilled water was used. ✓

Commented [A12]: You must describe how this was done. Was the surface seared with a scalding

metal knife before cutting out the sample with a sterile scalpel blade and transferring it to a sterile

Petri dish?

Organs affected by the hydatid cysts were collected and placed in bags and placed in refrigerated

containers and transported to the laboratory, and the heads were collected in the laboratory after

opening the water bags as shown in the paper.

Also, we used another method if the organs were infected with a few hydatid cysts with one cyst,

for example, the hydatid cyst was removed from the affected organ using a knife and kept in tubes

of 50 ml capacity

In the paper the authors state: “The protoscoleces were aseptically collected from infected liver and lung ..”

Their reply above does not explain how the samples were collected ‘aseptically’.

Aseptically: It was deleted from the paragraph and the method is explained in an easier way.

It means sterile method, that is, all tools used in the process of collecting and transporting samples

were clean and sterilized with 70% ethanol before use, in order to not contaminate the carcasses

during sampling.

My response: 70% alcohol is a disinfectant but does not sterilize. Autoclaving sterilizes as it kills all life forms and viruses. Alcohol does not. The method you describe is antiseptic.

Changes were made to the manuscript as per your directions

The protoscoleces were collected from infected livers and lungs of animals that contained hydatid cysts slaughtered at the Al-Kharj slaughterhouse in Saudi Arabia and transferred to the parasitology laboratory of the zoology department of the University of King Saud.

Commented [A16]: Most research projects use a 5% level of significance (P< 0.05), and far fewer use 1% (P< 0.01). Why did you use such a high level of significance (0.1%, P < 0.001)?

P < 0.001, it was correcting P< 0.01 indicates high significant differences, and this is what happened between the control and the highest concentration. While P< 0.05 indicates the presence of close or low significant differences between the low concentrations or with the control.

This is in the ‘Materials and Methods’ section so it seems the authors chose the high level of significance (P< 0.001). I advise moving it to the ‘Results’ section to avoid confusion.

Done ✓

Commented [A17]: You should explain in a "Key:" what is meant by "Appearance" and the

differences between "Medium, "Strong" and "Strong broad". You should also describe what you

mean by "Transmittance".

Medium, Strong, and Strong broad This expresses the number of active chemicals in the extract.

Some compounds may be present in greater quantity than others. while "Appearance" expresses

the presence of active chemicals in the extract

Does this response indicate the authors will include these explanations in the paper?

I think it is not necessary to add this clarification in the paper because the method is known and clear.

My response: It is not clear to me and so probably other readers.

The explanation will not be added in the manuscript but an answer to your question. It is intended to clarify the presence of active compounds in the extract.

Commented [A19]: You should state the levels of significance for both exposures.

The control group is the standard group that we rely on for comparison between the other experimental groups, and it is the group that did not add extracts or was not treated, and therefore the eggs will not die, and therefore the column will be at zero point, so it does not appear.

This response does not report the P levels for the exposures.

My response: No response from the authors.

Commented [A20]: The eggs in both A and B images are stained red. What color is the egg in A

supposed to be?

In picture(B), the eggs were not dyed in red color, as the egg wall was not dyed and was not affected compared to the other picture.

We note in the attached picture(B) in the paper that the outer layer of the eggs is irregular and we note that the red color inside the eggs, even the size of the eggs is larger than the eggs that are not dyed red, and this is evidence of the effect of the eggs.

As for the color of the eggs, we note that the natural color of the eggs is brown, as is the color in the other picture attached to the paper

As both eggs in Figure 2 A and B are red and the authors state the natural color is brown, the authors’ response does not make sense to me.

My response: The differences in egg size and thickness of the wall are obvious and although there are detectable differences in color intensity both eggs are still generally crimson in color so including an image of eggs before the test should indicate the changes that occur with death and therefore clarify the changes.

The pictures of the eggs were changed before the test and after the test in order to notice the changes on the eggs after exposure to the extract of grape leaves.

Figure 2. Effect of Vitis vinifera leaves extract on eggs viability. The unviable eggs after treatment with extract and staining with 0.1% eosin (A). The viable eggs did not change in color after staining with 0.1% eosin (B). Scale bar = 20 µm.

Commented [A22]: Don't you mean on viable eggs?

This refers to: “Line 210: Figure 4. The main effects of VVLE on unviable eggs at 24 and 48 hours of

exposure ..” As the project is investigating how effective VVLE is in killing Echinococcus eggs they must be

alive (viable) before exposure. If they are already dead (“unviable”) there is no point to the exposure.

The figure shows the effect of the extract on viable eggs and the amount of mortality in them after

exposure to the extract and its effect on them to make them non-viable eggs.

Yes, in the experiment, viable eggs were used, not non-viable eggs

It was modified in the scientific paper from non-viable eggs to viable eggs. ✓

Commented [A29]: What does this mean? The main surgical multiples of cystic echinococcosis are

relapsed, subaltern cystic echinococcosis, and anaphylactic shock

That is, a secondary infection occurs when the water sacs are opened and this fluid that contains

the heads leaks to other Organ, and therefore the surgery is considered dangerous in the occurrence

of a secondary infection. Also, surgery and opening the sacs cause great complications for those

infected with it.

Accepted, but I do not understand the use of “surgical multiples” or “subaltern” here.

It means that if the liquid in the bags containing the protoscolios leaks, the infection will occur again, and in this way, the infection continues in the animal, and this is considered one of the most important complications. Therefore, surgery is a problem, and it is preferable to use medicines or natural products that work to get rid of this parasite.

My response: I understand the risks posed by surgical procedures but the authors have not addressed my comments that I do not understand their use of the words “surgical multiples” or “subaltern”. According to dictionaries “subaltern” means lower military rank which does not make sense in the contextual use here.

This paragraph has been modified in the manuscript thus:

The main Surgical problems of cystic echinococcosis are relapse, subaltern cystic echinococcosis, and anaphylactic shock because of intraoperative rupture of the cyst and leakage of the contents of the cyst (protoscoleces), which is noticed in almost 10% of infected cases.

The authors replied to several comments that I do not understand what they meant with “The sentence in the paper has been modified.” I do not know how the paper was modified so I cannot determine whether it is now any clearer to readers.

Yes, your suggestions on the manuscript are modified to be clear to readers as directed
